# Prostate Cancer Lung Metastasis: Clinical Insights and Therapeutic Strategies

**DOI:** 10.3390/cancers16112080

**Published:** 2024-05-30

**Authors:** Ahmed M. Mahmoud, Amr Moustafa, Carter Day, Mohamed E. Ahmed, Wael Zeina, Usama M. Marzouk, Spyridon Basourakos, Rimki Haloi, Mindie Mahon, Miguel Muniz, Daniel S. Childs, Jacob J. Orme, Irbaz Bin Riaz, A. Tuba Kendi, Bradley J. Stish, Brian J. Davis, Eugene D. Kwon, Jack R. Andrews

**Affiliations:** 1Department of Urology, Mayo Clinic, Rochester, MN 55905, USA; mahmoud.ahmed@mayo.edu (A.M.M.);; 2Department of Internal Medicine, The Brooklyn Hospital Center, Brooklyn, NY 11201, USA; 3Department of Internal Medicine, Ain Shams University, Cairo 11566, Egypt; 4Department of Medical Oncology, Mayo Clinic, Rochester, MN 55905, USA; 5Department of Medical Oncology, Mayo Clinic, Scottsdale, AZ 85259, USA; 6Department of Radiology, Division of Nuclear Medicine, Mayo Clinic, Rochester, MN 55905, USA; kendi.ayse@mayo.edu; 7Department of Radiation Oncology, Mayo Clinic, Rochester, MN 55905, USA; 8Department of Urology, Mayo Clinic, Phoenix, AZ 85054, USA

**Keywords:** prostate cancer, lung metastasis, systemic therapy, metastasis-directed therapy

## Abstract

**Simple Summary:**

In our paper, we examine various facets of prostate cancer lung metastasis. We explore their clinical manifestations, such as respiratory symptoms and potential complications. Diagnostic modalities, including imaging techniques and biomarker analysis, are scrutinized for accurate identification. Treatment strategies, encompassing surgery, radiation therapy, chemotherapy, and targeted therapies, are discussed for their efficacy. Survival outcomes are assessed to gauge the effectiveness of different interventions. Through comprehensive analysis, we aim to provide insights into managing prostate cancer lung metastasis and enhancing patient care and outcomes.

**Abstract:**

Prostate cancer lung metastasis represents a clinical conundrum due to its implications for advanced disease progression and the complexities it introduces in treatment planning. As the disease progresses to distant sites such as the lung, the clinical management becomes increasingly intricate, requiring tailored therapeutic strategies to address the unique characteristics of metastatic lesions. This review seeks to synthesize the current state of knowledge surrounding prostate cancer metastasis to the lung, shedding light on the diverse array of clinical presentations encountered, ranging from subtle radiological findings to overt symptomatic manifestations. By examining the diagnostic modalities utilized in identifying this metastasis, including advanced imaging techniques and histopathological analyses, this review aims to provide insights into the diagnostic landscape and the challenges associated with accurately characterizing lung metastatic lesions in prostate cancer patients. Moreover, this review delves into the nuances of therapeutic interventions employed in managing prostate cancer lung metastasis, encompassing systemic treatments such as hormonal therapies and chemotherapy, as well as metastasis-directed therapies including surgery and radiotherapy.

## 1. Introduction

Prostate cancer (PCa) remains the most predominant and second-leading cause of cancer-related deaths among males, as well as attaining the status of being the primary cancer with a high incidence-to-mortality ratio in the US [1,2]. Despite the favorable control of the disease and survival outcomes following the radical prostatectomy for localized disease, distant metastasis ultimately occurs in 17%, and out of those patients, 20–30% acquire visceral metastasis [3,4,5]. Distant metastasis, particularly visceral metastasis, has a significant negative prognostic factor, resulting in poorer health-related quality of life and a rise in cancer-related mortalities [3,6,7]. Despite this, not all patients with visceral metastasis have a consistently severe disease progression. Indeed, patients with lung metastasis often have a more favorable course of disease than those with liver metastasis, and the prognosis is more akin to lymph node or bone metastasis [8,9].

As reported by autopsy studies, lung metastasis has been identified as the second most prevalent extranodal metastasis in PCa (46%) behind bone metastasis (90%) [10]. With different pathways of PCa dissemination, the most common pathway for lung involvement with PCa is through the Caval pathway, which signifies a critical juncture in the disease course, indicating both the aggressiveness of the cancer and its potential to infiltrate vital circulatory pathways. The inferior vena cava, as a major conduit for venous blood returning to the heart, becomes a conduit for cancer cells originating from the prostate gland. This infiltration not only poses challenges in terms of treatment but also raises concerns regarding the potential for systemic dissemination [10]. The lungs, being a common destination for metastatic spread due to their rich vascularization, become a site where these cancer cells may seed and form secondary tumors, further compromising respiratory function and overall health [10].

Over the past years, there has been a noticeable rise in the occurrence of lung metastasis stemming from prostate cancer (PCa) [11]. Enhanced imaging techniques such as C11-Choline and prostate-specific membrane antigen (PSMA) PET scans are believed to offer improved capabilities in detecting lung metastasis in the early stages. Additionally, the introduction of new systemic treatment options has contributed to enhanced longevity and improved survival outcomes for patients with PCa [12]. With different lines of treatment proposed for this group of patients, there are still various obstacles encountered in adequately understanding, diagnosing, and treating PCa patients with lung metastasis. Given the dearth of understanding about metastatic lung lesions in PCa, we decided to conduct a comprehensive review of the most recent evidence on the clinical presentations, diagnostic methods, therapeutic approaches, and prognosis of lung metastasis in PCa.

## 2. Materials and Methods

### 2.1. The Literature Search Strategy

We conducted a comprehensive literature search to identify relevant studies focusing on prostate cancer metastasis to the lung. Electronic databases including PubMed/MEDLINE, Embase, and Web of Science were searched up to 1 February 2024, without any language restrictions. The search strategy included a combination of keywords and medical subject headings (MeSH) terms related to prostate cancer lung metastasis such as “prostate cancer”, “metastatic prostate cancer”, “advanced prostate cancer”, “lung metastasis”, “lung neoplasm”, “pulmonary metastasis”, and “prostate lung metastasis”. Additionally, manual searches of reference lists from retrieved articles and relevant review papers were performed to ensure the inclusivity of the review. **Figure 1** presents the flowchart of our search through the literature.

### 2.2. Inclusion and Exclusion Criteria

Studies were included if they met the following criteria: (1) original research articles, case reports, or systematic review articles investigating prostate cancer metastasis to the lung; (2) studies reporting clinical characteristics, diagnostic modalities, treatment approaches, or outcomes related to prostate cancer lung metastasis; and (3) availability of full-text articles. Studies were excluded if they were duplicates, conference abstracts, editorials, letters, commentaries, or describing patients who had other primary malignancies.

### 2.3. Study Selection and Data Extraction

Two independent reviewers (A.M.M. and A.M.) screened the titles and abstracts of the identified records to assess their eligibility for inclusion. Discrepancies were resolved through discussion or consultation with a third reviewer (C.A.). Full-text articles of potentially eligible studies were retrieved and assessed for final inclusion based on predefined criteria. Data were extracted using a standardized data extraction form, including study characteristics (e.g., author, year of publication), patient demographics, primary tumor characteristics, diagnostic methods, treatment modalities, and outcomes related to prostate cancer lung metastasis.

## 3. Prostate Cancer Lung Metastasis Presentations

### 3.1. Clinical and Laboratories

PCa lung metastasis may manifest with symptoms that primarily result from the spread of cancer cells to the lungs. In the early stages, individuals may not experience noticeable symptoms, and the metastasis may be asymptomatic [13,14,15]; however, as the disease progresses and the metastatic lesions affect lung function and adjacent structures, various symptoms may emerge. Persistent coughing, often accompanied by the production of sputum, can develop as a consequence of parenchymal lung involvement. Shortness of breath, or dyspnea, may occur due to impaired respiratory function caused by the presence of metastatic nodules. Chest pain may be present, resulting from the irritation or compression of surrounding tissues by the growing metastasis. In some cases, hemoptysis—coughing up blood—may occur, though this is relatively uncommon. Generalized fatigue and weakness may be attributed to the overall burden of the disease on the body [16].

As in **Table 1**, most of the patients across all the studies were asymptomatic, while only 13 out of the 58 studies included reported variable clinical symptoms ranging from general symptoms such as hematuria, increased urinary frequency, and weight loss to pulmonary symptoms like cough, dyspnea, chest pain, and hemoptysis [17,18,19,20,21,22,23,24,25,26,27,28,29]. Several factors contributed to the presenting symptoms, including the number and location of the metastatic lesions. In terms of clinical characteristics for those patients, the median age was 69 years, with a median Gleason score and PSA at the time of lung metastasis diagnosis of 8 and 3.8 ng/mL, respectively.

### 3.2. Diagnostic Methods

Although the involvement of the lung by prostate cancer is infrequent compared with lymph nodes or bone, it represents a significant presentation, especially in patients with widespread metastatic disease. This situation carries substantial consequences for patient prognosis and overall survival. Hence, it is imperative to prioritize early diagnosis and intervention, particularly in patients with a heightened clinical suspicion of metastasis [72]. With different diagnostic methods of PCa lung metastasis, herein we have sorted and examined them distinctly, considering the specific characteristics of each diagnostic approach.

#### 3.2.1. Imaging Modalities

Various imaging techniques can be utilized to detect lung metastasis as mentioned in **Table 1**, including chest X-ray (CXR), computed tomography (CT) scan, magnetic resonance imaging (MRI), and positron emission tomography (PET) scan. The selection of the optimal imaging method depends on factors like the tumor’s biological behavior, the sensitivity and specificity of the imaging modality, considerations regarding radiation exposure, and cost-effectiveness. The decision is tailored to the specific clinical situation to ensure precise diagnosis and effective treatment planning. Lung metastasis in advanced prostate cancer often exhibits either a lymphangitic or nodular radiological profile. The higher occurrence of the lymphangitic pattern is attributed to the direct invasion of lung lymphatics, while the nodular pattern results from hematogenous spread [10].

Nodular involvement is the most common observation, whereas pulmonary lymphangitic involvement by prostate cancer is extremely rare, occurring in less than 0.2% of patients [73]. Lymphangitic involvement, termed pulmonary lymphangitic carcinomatosis, is pathologically defined by the presence of tumor thrombi within the lymphatic vessels of bronchovascular bundles, interlobular septa, and pleura. On imaging, it typically presents as multiple linear densities forming a reticular network with thickened and irregular broncho-vascular bundles. Another frequently observed radiographic presentation is the “tree-in-bud” pattern, which illustrates bronchiolar luminal impaction outlining the usually unseen peripheral airway branching [74,75,76]. Atypical PCa lung metastasis presentations were reported in the literature including cavitating, excavated, cystic lesions, and nodules with ground-glass appearance [27,77,78].

Historically, **chest X-rays** were used as the primary imaging modality for detecting lung metastasis in PCa. On X-rays, the lung metastasis usually appears as multiple round or oval-shaped nodules with calcification in some cases. However, X-rays have limitations in detecting smaller lesions and providing detailed information [26,62,63,65,67,70]. Even though, in some cases, chest X-ray may be part of the initial diagnostic workup or used for monitoring purposes. It can offer a broad overview of the lungs and may detect larger or more advanced lung metastasis [79].

Therefore, more advanced imaging techniques, such as CT scans and MRI, are generally preferred for evaluating the presence of metastasis in the lungs. The **CT scan** is the most sensitive imaging method for detecting pulmonary metastasis due to its superior spatial and contrast resolution and the absence of overlap with adjacent structures like bones and vessels [80]. In comparison to chest X-ray (CXR), a CT scan can identify a greater number of nodules, including those smaller than 5 mm, and can detect approximately three times as many noncalcified nodules. Additionally, CT scans can reveal various findings, such as lymphadenopathy, involvement of the pleura, chest wall, airways, and blood vessels, as well as abnormalities in the upper abdomen and bones, all of which may have an impact on treatment decisions [81].

PCa lung metastasis can appear differently on CT chest and MRI images. In a CT chest scan, metastatic lesions may present as nodules or masses within the lung tissue; these nodules can vary in size and density [79]. On **MRI** images of the chest, metastatic lesions may appear as areas with abnormal signal intensity. MRI is particularly useful for soft tissue imaging as it can provide detailed information about the anatomy and characteristics of lesions [82]. It is important to note that the appearance of prostate cancer lung metastasis on imaging studies can be diverse, and the interpretation may depend on various factors such as the size and location of the lesions, as well as the specific imaging techniques used [82].

Nowadays, **PET scans** play a significant role in the assessment of PCa patients in general and lung metastasis specifically, offering valuable insights into the metabolic activity of tissues. Unlike anatomical imaging techniques such as CT or MRI, PET scans provide functional information by detecting areas of increased metabolism, which aids in distinguishing between benign and malignant lesions and is particularly advantageous for detecting widespread or multifocal metastasis [83]. The use of PET scans in conjunction with CT scans, known as PET/CT fusion imaging, further enhances diagnostic accuracy by combining anatomical and functional information. This hybrid imaging approach allows for precise localization of hypermetabolic lesions within the lung tissue and facilitates a comprehensive assessment of the extent and distribution of metastasis. Additionally, PET scans can be instrumental in monitoring treatment response and detecting disease recurrence, aiding clinicians in making informed decisions regarding the management of lung metastasis [83].

Despite the advantages of PET scans, few research articles investigate the use of Ga-PSMA PET/CT in evaluating lung metastasis in prostate cancer (PCa). Damjanovic et al. (2019) conducted a study encompassing 739 PCa patients, revealing 91 confirmed lung metastases in 20 individuals. Their findings demonstrated that 72.5% of this metastasis exhibited PSMA-positive characteristics, while 27.5% were PSMA-negative. Despite the prevalent PSMA expression in most lung metastasis, a significant subset displayed PSMA negativity. Additionally, benign pulmonary lesions exhibited a moderate tracer uptake, lower than that seen in PSMA-positive lung metastasis but higher than PSMA-negative ones. Consequently, the sole reliance on the SUVmax of 68Ga-PSMA PET proved inadequate for distinguishing pulmonary metastasis from benign lung lesions [84]. This conclusion corresponds with the outcomes of Pyka et al., which similarly emphasized the difficulty in discerning lung metastasis of PCa from primary lung cancer based on the SUVmax values of 68Ga-PSMA PET [85].

#### 3.2.2. Histopathology

Histopathological analysis of lung metastasis in prostate cancer provides invaluable insights into the biological behavior and characteristics of the metastatic lesions. PCa cells in lung metastasis often retain the glandular features seen in the primary tumor, aiding in their identification. The majority of lung metastasis in prostate cancer arises from prostate adenocarcinoma, given its status as the most prevalent histological subtype of PCa [14,15,33,38]. Nevertheless, less common histological subtypes, such as small cell carcinoma and neuroendocrine carcinoma of the prostate, have been documented [32,47,50,54]. In addition to its role in tumor staging, histology plays a significant role in understanding the dissemination of prostate cancer to the lungs. Hematoxylin and eosin staining of lung metastasis in prostate adenocarcinoma reveal glandular patterns, a characteristic further confirmed by positive prostate-specific antigen (PSA) staining in immunohistochemistry [16].

The histopathological examination may also reveal the extent of differentiation and the presence of any unique markers, contributing to a more comprehensive understanding of the tumor’s molecular profile. This detailed examination is crucial for accurate diagnosis, determining the appropriate course of treatment, and assessing the potential impact on patient prognosis. Additionally, histopathology plays a pivotal role in guiding personalized therapeutic approaches and advancing our knowledge of the complex nature of prostate cancer metastasis to the lungs. In light of our research findings, which indicate that ultrasound or CT-guided biopsy and tissue evaluation were commonly employed in most of the studies, it can be inferred that histopathology stands out as the most accurate method for diagnosing lung metastasis in PCa [12,13,21].

#### 3.2.3. Molecular and Genetics

The molecular and genetic landscape of prostate cancer lung metastasis is intricate, reflecting the heterogeneity of the disease. Recent advancements in genomic profiling have unveiled a spectrum of genetic alterations that influence the evolution of metastatic lesions. Dysregulation of genes involved in DNA repair mechanisms, cell cycle control, and chromatin modification pathways are among the molecular events associated with prostate cancer metastasis to the lungs [86]. Notably, studies have highlighted the role of specific genetic markers, including TMPRSS2-ERG rearrangements and alterations in the TP53 gene, in driving the metastatic potential of prostate cancer cells [87,88]. The evolving field of somatic cancer genotyping plays a pivotal role in categorizing metastatic diseases, aiding in the differentiation of aggressive and indolent phenotypes [86,89]. Specific genomic alterations, notably compound disruptions of tumor suppressor genes or deficiencies in DNA damage repair genes, are frequently observed in prostate cancers with high-risk clinical features [90].

Historically, the genomic features of lung metastasis were poorly explored in the literature apart from two case series that investigated the genomics of lung metastasis in metastatic hormone-sensitive PCa (mHSPC) patients [87,91]. Shenderoev et al. identified 16 mHSPC patients with plenty of variations in mismatch repair (MMR) genes, homologous recombination deficit (HRD) genes, PI3K pathway genes, Wnt signaling pathway genes, and TP53 mutations [87]. Another study by Fonseca et al. observed a potential increase in changes affecting PTEN and the Wnt/β-catenin signaling genes RNF43 and APC, which underscores the necessity for further studies into the cellular pathways facilitating the spread of metastasis in the lung. Additionally, in some rare cases, the immunohistochemical assessment may reveal negative results for PSA, NKX3.1, and p501s, which are commonly used biomarkers for prostatic adenocarcinoma and showed a positivity for synaptophysin and chromogranin A that indicates adenocarcinoma of the prostate with neuroendocrine differentiation [92].

The identification of these molecular and genetic signatures not only contributes to a deeper understanding of the disease’s biology but also holds promise for the development of targeted therapies tailored to the genetic makeup of individual tumors. The evolving field of targeted therapies seeks to capitalize on these molecular insights to design more effective and personalized treatment strategies for PCa patients with lung metastasis [91].

## 4. Treatment Approaches

The management of PCa lung metastasis necessitates a comprehensive approach tailored to individual patient characteristics and disease factors. As in **Table 1**, treatment options typically involve systemic therapies, such as androgen deprivation therapy (ADT), androgen receptor pathway inhibitor (ARPI), chemotherapy, and radioligand drug, along with metastasis-directed therapies that aim at alleviating symptoms and controlling disease progression.

### 4.1. Systemic Therapies

**Androgen deprivation therapy** (**ADT**) serves as a pivotal component in managing lung metastasis stemming from prostate cancer, offering a multifaceted approach to impede disease progression. By curtailing the production of testosterone and other androgens, ADT exerts palliative effects, effectively slowing the advancement of metastatic prostate cancer. This therapeutic strategy remains a cornerstone in the treatment arsenal, contributing significantly to improving patient outcomes [12,15,17,18,20,21,23,25,28,29,30,35,37,43,44,45,49,54,55,56,59,61,63,64,65,68,71]. Another class is the **ARPIs** represent promising therapeutic options in the management of prostate cancer lung metastasis, offering targeted inhibition of the androgen receptor pathway to impede disease progression and alleviate symptoms [29,35]. ARPIs disrupt the signaling cascade essential for prostate cancer cell growth and survival, thereby exerting a profound anti-tumor effect. Clinical trials and real-world evidence have demonstrated the efficacy of ARPI in delaying disease progression and prolonging overall survival in patients with metastatic prostate cancer, including those with lung involvement [72,93]. By specifically targeting the androgen receptor pathway, ARPI addresses the underlying molecular drivers of prostate cancer metastasis, offering a targeted and personalized approach to treatment. Its favorable safety profile and oral administration further enhance its appeal as a therapeutic option for patients with prostate cancer lung metastasis, underscoring its role as a cornerstone in the evolving landscape of prostate cancer management [94,95].

**Chemotherapy** plays a significant role in the management of prostate cancer lung metastasis, particularly in cases of castration-resistant prostate cancer (CRPC) where other treatments may have limited efficacy. Chemotherapeutic agents such as docetaxel and cabazitaxel are commonly used to target rapidly dividing cancer cells and inhibit tumor growth [96]. These agents are administered either alone or in combination with other treatments to maximize their effectiveness. While chemotherapy may not be curative, it can help alleviate symptoms and prolong survival in patients with metastatic prostate cancer, including those with lung involvement [96].

**177Lu-PSMA therapy** represents a promising and evolving radioligand treatment approach for patients with prostate cancer lung metastasis, offering targeted radiation therapy directed at prostate-specific membrane antigen (PSMA)-expressing tumor cells [97]. This innovative therapy utilizes radiolabeled PSMA ligands to deliver therapeutic doses of beta radiation selectively to prostate cancer cells while sparing surrounding healthy tissues. By targeting PSMA, which is highly expressed in prostate cancer cells, 177Lu-PSMA therapy holds the potential to effectively eradicate metastatic lesions in the lungs, offering a personalized and precise treatment option [97,98]. Several studies have demonstrated encouraging results, with significant reductions in PSA levels, tumor burden, and in some cases a complete regression of multiple lung lesions as reported by Zhang et al. [98,99,100,101]. While further research is needed to optimize treatment protocols and assess long-term outcomes, 177Lu-PSMA therapy represents a promising therapeutic strategy in the management of prostate cancer lung metastasis, offering hope for improved outcomes for affected patients [97,101].

### 4.2. Metastasis-Directed Therapies (MDT)

Localized approaches such as radiation therapy or surgical resection may also be considered for palliation of symptoms or to address oligometastatic disease. The choice of treatment modality depends on various factors, including the extent of metastatic spread, the aggressiveness of the disease, the patient’s overall health status, and individual treatment goals [30]. Multidisciplinary collaboration among oncologists, urologists, radiation oncologists, thoracic surgeons, and other specialists is crucial in formulating personalized treatment plans that optimize survival outcomes for patients with prostate cancer lung metastasis.

**Surgical resection** of prostate cancer lung metastasis remains a controversial yet potentially beneficial option in select cases, particularly when metastases are localized and deemed resectable [30,52]. While systemic therapies such as hormone therapy and chemotherapy are often the mainstays of treatment for metastatic prostate cancer, surgical intervention may be considered for patients with limited metastatic burden and good overall health. The goal of surgical resection is to remove visible metastatic lesions from the lungs, potentially providing symptomatic relief, prolonging progression-free survival, and offering a chance for cure in some cases [52]. In a retrospective cohort study, surgical resection either alone or combined with systemic therapy was linked with favorable survival outcomes compared with systemic therapy alone [30]. However, the decision to proceed with surgery requires careful consideration of various factors including the location and size of metastatic lesions, the extent of metastatic disease, the patient’s general health status, and their willingness to undergo surgery [58]. Additionally, surgical resection is typically reserved for patients with oligometastatic disease, where metastasis is confined to a limited number of sites [58]. While surgical resection of prostate cancer lung metastasis may offer potential benefits, including improved quality of life and prolonged survival, it is essential to weigh the risks and benefits carefully and to individualize treatment decisions based on each patient’s unique circumstances [52,55].

**Radiotherapy** emerges as a vital component in the multimodal approach to managing prostate cancer lung metastasis, offering targeted and localized treatment to control disease progression [102]. With advancements in radiotherapy techniques such as intensity-modulated radiation therapy (IMRT) and image-guided radiotherapy (IGRT), precise delivery of radiation to metastatic lesions in the lungs is achievable while minimizing toxicity to surrounding healthy tissues. Moreover, emerging evidence supports the efficacy of stereotactic ablative radiotherapy (SABR), also known as stereotactic body radiotherapy (SBRT), in delivering high doses of radiation in a limited number of treatment sessions, thereby providing a convenient and effective treatment option for patients [52,58,100,101,102,103,104]. By delivering focused radiation to metastatic lesions, several studies showed that radiotherapy can induce tumor regression, improve local control, and complete resolution for patients with prostate cancer lung metastasis [12,15,24,29,38,70,71,102,103,104].

## 5. Prognosis and Oncological Outcomes

Metastasis to visceral organs in prostate cancer (PCa) typically indicates unfavorable disease outcomes [3,105]. Nevertheless, the clinical progression of patients with visceral metastasis can vary significantly. Specifically, patients with lung metastasis tend to experience a more favorable disease course compared to other sites of visceral metastasis [8]. As in **Table 1**, prognosis and survival outcomes in prostate cancer lung metastasis vary significantly according to the present studies, with survival ranging from 3 to 88 months and a median survival of 26 months [13,15,17,18,20,21,23,24,25,28,29,35,36,37,38,41,44,45,46,47,49,52,54,55,56,58,59,60,61,62,63,64,65,66,69,71]. The survival outcomes depend on several factors, including the extent of metastatic spread, the aggressiveness of the disease, the presence of comorbidities, and the effectiveness of treatment interventions [93].

Generally, prostate cancer lung metastasis is associated with a poorer prognosis compared to localized disease, reflecting the advanced stage of cancer and the challenges in managing metastatic spread [72,93]; however, with advancements in treatment modalities such as systemic therapies and MDT, there has been a notable improvement in survival outcomes for those patients [30,72,93]. Despite these advancements, prognosis remains variable and individualized treatment plans tailored to each patient’s specific circumstances are crucial in optimizing outcomes [30]. Multidisciplinary collaboration among healthcare providers is essential in providing comprehensive care and support to patients with prostate cancer lung metastasis, with the ultimate goal of improving prognosis and enhancing overall survival.

## 6. Conclusions

Prostate cancer lung metastasis represents a significant clinical challenge, often indicating advanced disease progression and posing therapeutic dilemmas for clinicians. Through this comprehensive review, we have synthesized current evidence regarding the wide variety of clinical presentations, diagnostic methods, therapeutic approaches, and prognosis associated with prostate cancer metastasizing to the lung. With the complexity of prostate cancer lung metastasis presentations, continued research efforts aimed at elucidating underlying mechanisms, identifying novel therapeutic targets, and optimizing treatment strategies are paramount. Furthermore, interdisciplinary collaboration and patient-centered care are essential in navigating the complexities of managing these metastatic lesions and improving the quality of life and survival outcomes of those patients.

## Figures and Tables

**Figure 1 cancers-16-02080-f001:**
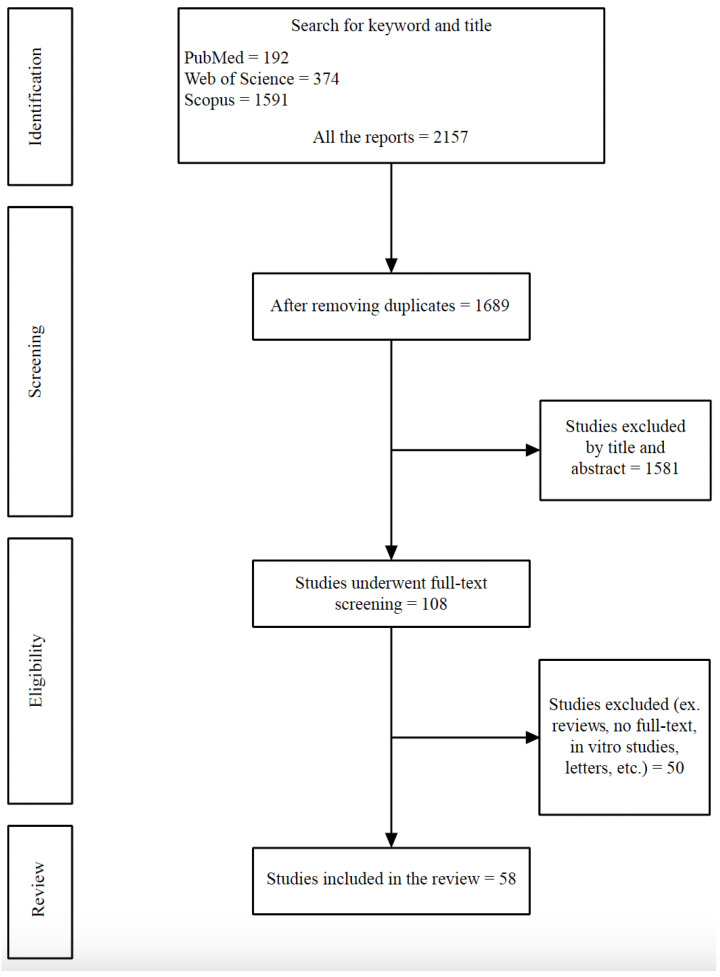
Flow diagram of the study selection.

**Table 1 cancers-16-02080-t001:** Patients’ characteristics from studies reported lung metastasis in prostate cancer.

Articles	Year of Publication	Type of Study	Number of Cases	Age/Median	Pathology	Gleason Score/Median	PSA/Median	Symptoms	Lines of Treatment	Prognosis	Survival (Months)	Methods of Diagnosing	Number of Met	Met’s Location	Concomitant Met
Mahmoud, et al. [30]	2023	Retrospective	75	69	Adeno	7	4	N/A	ADT, Surgical resection, Chemo	72% Survived	N/A	Chest CT, PET/CT	Multiple	Bilateral	N/A
Ceylan, et al. [13]	2023	Case report	2	57.11	Adeno	N/A	N/A	Asymptomatic	Surgical resection	Died	6	Chest CT, PET/CT	N/A	N/A	N/A
Kase, et al. [31]	2022	Case series	8	N/A	Adeno	7	3.1	N/A	ADT, Surgical resection	Survived	64	Chest CT	Multiple	Bilateral	N/A
Delbare, et al. [14]	2022	Case report	1	75	Adeno	7	6	Asymptomatic	N/A	N/A	N/A	Chest CT, F18-PSMA-PET/CT	Multiple	Bilateral	L.N
Kaneko, et al. [32]	2022	Case report	1	79	SCC	N/A	0.12	Asymptomatic	Chemo	Died	N/A	Chest CT, SCC antigen	Multiple	Bilateral	Liver, L.N
Kosaka, et al. [15]	2022	Case report	1	61	Adeno	7	0.76	Asymptomatic	ADT, RT, Surgical resection	Survived	84	Chest CT, PET/CT	Multiple	Bilateral	None
Cui, et al. [18]	2021	Case report	1	72	Adeno	8	12.64	Frequent urination	ADT	Survived	12	Chest CT	Solitary	Right	None
Tarabaih, et al. [33]	2021	Case report	1	70	Adeno	7	1.97	Asymptomatic	Surgical resection	Survived	N/A	Chest CT	Solitary	Right base	None
Yoshitake, et al. [34]	2021	Case report	1	83	Adeno	5	N/A	Asymptomatic	Surgical resection	Survived	N/A	Chest CT	2	Bilateral	None
Carrilho Vaz, et al. [35]	2020	Case reports	4	77, 76, 76, 75	Adeno	N/A, N/A, 7, N/A	1.69, 7.55, 6.36, 2.6	N/A	ADT, ARPI, Chemo, Surgical resection	Survived	N/A, 4, 9, N/A	Ga-PSMA PET/CT	1- Multiple, 2-Solitary, 3- Multiple, 4-Solitary	1- Bilateral, 2- Left upper, 3- Bilateral, 4- Right	N/A
Izawa, et al. [36]	2020	Case report	1	62	Adeno	9	4.25	Asymptomatic	VATS, Chemo	Survived	10	Chest CT	Multiple	Bilateral	None
Tang, et al. [28]	2020	Case report	1	48	Adeno	8	3.03	Hematuria	ADT, Chemo	Survived	19	Chest CT, PET/CT	2	Right	None
Tsakiridis, et al. [29]	2020	Case report	1	75	Adeno	7	5	Shortness of breath	ADT, ARPI, SBRT	Survived	48	Chest CT, 18F-NaF-PET/CT, 18F-DCFPyL PET/CT	2	Left lower, Left hilar	None
Polistina, et al. [27]	2020	Case report	1	74	Adeno	9	N/A	Weight loss	N/A	N/A	N/A	Chest CT, 18F PET/CT, FDG PET/CT	Multiple	Bilateral	N/A
Wu, et al. [37]	2020	Case report	1	74	Adeno	8	2	Asymptomatic	ADT	Survived	36	Chest CT	Solitary	Right	N/A
Ciriaco, et al. [38]	2019	Case series	9	61	Adeno	8	1.66	Asymptomatic	RT, Surgical resection	Survived	24.5	Ga-PSMA PET/CT	4 Solitary, 5 multiple	Left, Right	None
Mosca, et al. [25]	2019	Case report	1	63	Adeno	8	1.32	Cough	ADT, Surgical resection	Survived	32	Chest CT, PET/CT	2	Left	None
Katsui, et al. [39]	2019	Case report	1	62	Adeno	8	N/A	N/A	N/A	N/A	N/A	N/A	N/A	N/A	N/A
Seniaray, et al. [40]	2019	Case report	1	63	Adeno	8	189.2	Asymptomatic	N/A	N/A	N/A	Ga PSMA PET/CT	Multiple	Bilateral	L.N
Boschian, et al. [41]	2018	Case report	1	69	Adeno	7	0.4	Asymptomatic	Surgical resection	Survived	36	FDG-PET/CT	Solitary	Left	N/A
Damjanovic, et al. [12]	2018	Retrospective	34	70.6	Adeno	9	123.6	N/A	ADT, Chemo, RT	N/A	N/A	Ga-PSMA-PET/CT	Multiple	Bilateral	N/A
Polverari, et al. [42]	2018	Case report	1	78	Adeno	7	0.3	Asymptomatic	Surgical resection	N/A	N/A	68Ga-PSMA-11 PET/CT, FDG PET/CT	Solitary	Upper right	None
Reinstatler, et al. [43]	2018	Case report	1	60	Adeno	8	44	Asymptomatic	ADT, Surgical resection, Chemo	N/A	N/A	Chest CT	Multiple	Left	L.N
Hokamp, et al. [44]	2017	Case report	1	63	Adeno	N/A	1.60	Asymptomatic	ADT, Surgical resection	Survived	N/A	68Ga PSMA PET/CT	Solitary	Right	L.N
Mortier, et al. [45]	2017	Case report	1	82	Adeno	6	3.32	Asymptomatic	ADT, Surgical resection	Survived	12	Chest CT, PET/CT	Solitary	Right	N/A
Rush, et al. [46]	2017	Case report	1	70	Adeno	8	2.9	Asymptomatic	Surgical resection	Survived	24	Chest CT, PET-CT	Solitary	Left	N/A
Gago, et al. [21]	2016	Case reports	3	63, 62, 79	Adeno	9, 7, 7	12.3, N/A, 2	Cough	ADT, Surgical resection, Chemo	Died, Survived, Survived	24, 60, 5	Chest CT, PET/CT, Endobronchial US	Multiple	Right, Left, Bilateral	None
Geraldo, et al. [47]	2016	Case report	1	60	Sarcomatoid	8	4.63	Asymptomatic	Surgical resection	Survived	12	Ga-PSMA PET/CT, F-FDG PET/CT	Solitary	Left	L.N
Hung-Yi Su, et al. [48]	2016	Case report	1	54	Adeno	7	11.08	Asymptomatic	Surgical resection	Survived	N/A	F-FDG PET/CT	3	Right, mediastinum	L.N
Erdogan, et al. [19]	2015	Case report	1	71	Adeno	7	3.83	Chest pain	Surgical resection	N/A	N/A	FDG PET/CT	Solitary	Right	Bone
Kamiyama, et al. [49]	2015	Case report	2	59, 69	Adeno	8, 9	0, 5.6	Asymptomatic	ADT, Chemo	Died	6	Chest CT	Multiple	N/A	L.N, Bone, Brain
Maebayashi, et al. [24]	2015	Case report	1	50	Adeno	9	N/A	Bloody sputum	Chemo, RT, Surgical resection	Died	30	Chest CT, PET/CT	Solitary	Left	Systemic
Fukuoka, et al. [20]	2014	Case report	1	87	Adeno	7	66.6	Cough	ADT	Died	18	CXR, Chest-CT	Multiple	Bilateral	Liver
Treglia, et al. [50]	2013	Case report	1	68	Neuroendocrine	N/A	N/A	N/A	Surgical resection	N/A	N/A	PET/CT	N/A	N/A	N/A
Pepe, et al. [51]	2012	Case report	1	75	Adeno	7	0	Asymptomatic	Surgical resection	Survived	N/A	FDG PET/CT	N/A	N/A	N/A
Wallis, et al. [52]	2011	Case report	1	53	Adeno	9	N/A	Asymptomatic	Surgical resection	Survived	12	Chest CT, PET/CT	Multiple	Right	None
Sakai, et al. [53]	2010	Case report	1	74	Adeno		N/A	N/A	Surgical resection	N/A	N/A	N/A	N/A	N/A	N/A
Goto, et al. [54]	2010	Case report	1	73	Sarcomatoid	9	N/A	Asymptomatic	ADT, Surgical resection	Survived	10	Chest CT, Bronchoscopy, PET scan	Solitary	Right	None
Khandani, et al. [22]	2009	Case report	1	78	Adeno	8	8.5	Chest pain and hemoptysis	Surgical resection	Survived	N/A	Chest CT, FDG PET/CT	Solitary	Left	Subcarinal L.N
Boyer, et al. [55]	2009	Case report	1	65	Adeno	9	10.56	Asymptomatic	ADT, Surgical resection	Survived	11	Chest CT	Solitary	Upper left	N/A
Pruthi, et al. [56]	2007	Case report	1	72	Adeno	6	1.9	Asymptomatic	ADT, Surgical resection	Survived	36	Chest CT, PET CT, Bone scan	Solitary	Right	None
Maeda, et al. [57]	2006	Case report	1	71	Adeno	N/A	4	Asymptomatic	Surgical resection	Survived	N/A	Carbon-11 Choline PET-/CT	Solitary	Left upper	None
Kirby, et al. [23]	2005	Case report	1	59	Adeno	9	23	Dyspnea and hemoptysis	ADT	Survived	7	Chest CT	Multiple	Right	None
Chao, et al. [17]	2004	Case report	1	68	Adeno	9	0.4	Cough and dyspnea	ADT, Surgical resection	Survived	144	CXR	Solitary	Left lower	None
Hofland, et al. [58]	2000	Case report	1	49	Adeno	9	1	Asymptomatic	Surgical resection	Lost to follow-up	10	CXR, Chest CT	Solitary	Left lower	L.N
Kume, et al. [59]	1999	Case report	2	56	N/A	N/A	N/A	N/A	Bilateral orchiectomy, ADT	Survived	88, 32	CXR, Chest CT	Multiple	Bilateral	None, Bone
Smith, et al. [60]	1999	Case report	1	70	Adeno	9	2.1	Asymptomatic	Surgical resection	Survived	12	CXR, Chest CT	Solitary	Right lower	None
Behrakis, et al. [61]	1997	Case report	1	71	N/A	N/A	N/A	N/A	ADT	Survived	8	CXR	Multiple	N/A	N/A
Allen, et al. [62]	1996	Case report	1	59	Adeno	N/A	N/A	N/A	Bilateral orchiectomy	Survived	3	CXR	Multiple	Bilateral	None
Harris, et al. [63]	1996	Case report	1	76	Adeno	N/A	42.6	Asymptomatic	ADT	Survived	60	CXR	Multiple	Bilateral	None
Leibman, et al. [64]	1995	Case report	1	78	Adeno	7	0.4	Asymptomatic	ADT, Chemo	Died	20	CXR, Chest CT	3	2 Right, 1 Left	Brain
Fabozzi, et al. [65]	1995	Retrospective	47	N/A	Adeno	7	N/A	N/A	ADT, Orchiectomy, Chemotherapy	N/A	HSPC: 25, CRPC: 13	CXR	N/A	N/A	N/A
Cusan, et al. [66]	1994	Case report	1	60	Adeno	N/A	N/A	Asymptomatic	Flutamide, thoracotomy	Survived	25	CXR, Chest CT	Multiple	Bilateral	None
Eastham, et al. [67]	1993	Case report	1	69	Adeno	7	N/A	Asymptomatic	Bilateral orchiectomy	Survived	N/A	CXR	Multiple	Bilateral	None
Rockey, et al. [68]	1990	Case report	1	83	Adeno	N/A	N/A	N/A	ADT	N/A	N/A	Chest CT	Solitary	Left	None
Bromberg, et al. [69]	1989	Case report	1	N/A	N/A	6	N/A	N/A	Bilateral orchiectomy, Wedge resection	Survived	13	N/A	N/A	N/A	N/A
Petras, et al. [26]	1983	Case report	1	59	N/A	N/A	N/A	Cough	Bilateral orchiectomy	Survived	N/A	CXR, Bone scan	Multiple	Bilateral	Bone
Panella, et al. [70]	1980	Case report	1	76	Adeno	N/A	N/A	N/A	RT, bilateral orchiectomy, chemo	N/A	N/A	CXR, transthoracic needle biopsy	Multiple	Bilateral	N/A
Varkarakis, et al. [71]	1974	Retrospective	26	64	N/A	N/A	N/A	N/A	ADT, RT	Died	14.9	CXR	N/A	N/A	Bone

Abbreviations: PSA: Prostate-specific antigen; Met: metastasis; Adeno: Adenocarcinoma; N/A: not available; ADT: androgen deprivation therapy; PET: positron emission tomography; CT: computed tomography; SCC: small cell carcinoma; L.N: lymph node; PSMA: prostate-specific membrane antigen; ARPI: androgen receptor pathway inhibitors; VATS: video-assisted thoracic surgery; SBRT: stereotactic body radiation therapy; RT: radiotherapy; CXR: chest X-ray; HSPC: hormone-sensitive prostate cancer; CRPC: castration-resistant prostate cancer.

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
