# Peer review of "Prostate Cancer Lung Metastasis: Clinical Insights and Therapeutic Strategies"

_cancers, 2024, doi:10.3390/cancers16112080_

Round 1

Reviewer 1 Report

Comments and Suggestions for Authors

I would like to congratulate the authors on a well written and thought-out review paper which covered the subject effectively. Although the review does not present anything “new” the approach and extent of the review makes it a valuable paper. There are a few minor comments.

1)      Many times, throughout the manuscript the authors mention personalized medicine but do not go into detail. I think the addition of more information concerning personalized medicine would improve the paper.

2)      I do not know how much information exists regarding new advanced surgical approaches for the treatment of metastatic prostate cancer, but if there is any this should be included.

3)      Many times throughout the paper the topic of the paragraph or section is bold- Is this necessary?

4)      Line 43 “as well as, it has attained the status”- -as well as attaining the status.

5)      Line 45- remove the

6)      Line 53- lung metastasis have been identified- lung metastasis has been identified or metastases.

7)      Line s 125 ND 131- The statements are grammatically correct but sound clumsy- I would suggest rewording them.

Author Response

  1. Many times, throughout the manuscript the authors mention personalized medicine but do not go into detail. I think the addition of more information concerning personalized medicine would improve the paper.

We appreciate the reviewer’s comment, and unfortunately, prostate cancer (PCa) lung metastasis has not been extensively studied in the literature, resulting in a significant knowledge gap in this area. In our review, we endeavored to thoroughly explore all available treatment modalities for these patients, providing detailed descriptions of each approach and emphasizing the importance of personalized treatment which is missing in the current time. Given the absence of well-established guidelines for this patient group, further investigation is imperative to determine the most effective treatment protocols.

  1. I do not know how much information exists regarding new advanced surgical approaches for the treatment of metastatic prostate cancer, but if there is any this should be included.

Thank you very much for bringing this point up. Apart from a limited number of case reports and retrospective studies, surgical approaches for managing prostate cancer (PCa) lung metastasis have not been heavily investigated, leading to a significant scarcity of data. This gap in the literature underscores the imperative need for further randomized clinical trials. Such trials are essential to advance our understanding of the efficacy and safety of surgical interventions for this specific patient group, ultimately guiding evidence-based clinical practice and improving patient outcomes.

  1. Many times, throughout the paper the topic of the paragraph or section is bold- Is this necessary?

This is a good question. In addition to ensuring that tables and figures are presented in bold for emphasis, we have strategically highlighted key points in bold. These include diagnostic methods such as CT, MRI, and PET, as well as treatment lines. This formatting choice is intended to provide clear reference points for readers, thereby enhancing the navigability of the manuscript and ensuring that critical information is readily accessible.

  1. Line 43 “as well as, it has attained the status”- -as well as attaining the status.

Thank you for your suggestion, we modified the text.

  1. Line 45- remove the

Thank you, we removed “the” from this sentence.

  1. Line 53- lung metastasis have been identified- lung metastasis has been identified or metastases.

We appreciate this note and correct this error.

  1. Lines 125 AND 131- The statements are grammatically correct but sound clumsy- I would suggest rewording them.

We modified the text according to the reviewer’s comment. 

Reviewer 2 Report

Comments and Suggestions for Authors

a comprehensive review article on lung metastasis in PCa

Title - should include "narrative review" - Major

abstract - clearly resuming the essence of the manuscript - no remarks

Introduction - concise description of the problem and the current state of knowledge on it - no remarks

material and methods - clearly describing the process of reviewing - no remarks

paragraph 3 and 4 - comprehensive review on diagnostics and treatment of lung metastasis in resent day - no remarks

5. Prognosis and Oncological Outcomes - 

table 1 is sub-standard - needs better formatting - unreadable at this time - Major

Conclusions - nicely substantiated

Author Response

- Title - should include "narrative review" - Major

We hold the reviewer's suggestion in high regard and sincerely appreciate the recommendation. Nevertheless, it is important to clarify that the inclusion of the phrase "narrative review" is not universally mandated for all review articles. We are of the considered opinion that the existing title is more engaging and appealing to our target audience, thereby enhancing the article's impact and accessibility. By maintaining the current title, we aim to attract a broader readership and foster greater interest in the subject matter.

- Table 1 is sub-standard - needs better formatting - unreadable at this time - Major

Thank you very much for highlighting this point. In response, we have adjusted the font size of the table to enhance its readability. This formatting approach has been previously employed in a systematic review for prostate cancer (PCa) brain metastasis (PMID: 36474619). By adopting this formatting style, we aim to ensure the clarity of the data presented, thereby facilitating a more comprehensive understanding for our readers for every single aspect of PCa lung metastasis.

- Abstract - clearly resuming the essence of the manuscript - no remarks

- Introduction - concise description of the problem and the current state of knowledge on it - no remarks

- Material and methods - clearly describing the process of reviewing - no remarks

- Paragraph 3 and 4 - comprehensive review on diagnostics and treatment of lung metastasis in resent day - no remarks

- Prognosis and Oncological Outcomes - 

- Conclusions - nicely substantiated

We truly appreciate the reviewer’s positive feedback regarding our manuscript.

Reviewer 3 Report

Comments and Suggestions for Authors

This is an interesting review on lung metastases in prostate cancer patients.

Concerning treatment options, it should maybe be mentionned that patients with visceral mets were excluded from COUGAR-302

Author Response

- Concerning treatment options, it should maybe be mentioned that patients with visceral mets were excluded from COUGAR-302

We appreciate the reviewer’s insightful comment regarding the exclusion of prostate cancer visceral metastases from COUGAR-302 trial. Indeed, the omission of this patient population mostly due to the poor survival outcomes from numerous clinical studies has created a significant gap in the literature. However, the primary objective of our review is to address this gap by focusing specifically on the clinical presentation, diagnostic challenges, and treatment strategies associated with lung metastases in prostate cancer.

Round 2

Reviewer 2 Report

Comments and Suggestions for Authors

The authors had taken into account all the reviewer's recommendations.

The manuscript can be published in its present form